# Insect Transcription Factors: A Landscape of Their Structures and Biological Functions in *Drosophila* and beyond

**DOI:** 10.3390/ijms19113691

**Published:** 2018-11-21

**Authors:** Zhaojiang Guo, Jianying Qin, Xiaomao Zhou, Youjun Zhang

**Affiliations:** 1Department of Plant Protection, Institute of Vegetables and Flowers, Chinese Academy of Agricultural Sciences, Beijing 100081, China; guozhaojiang@caas.cn (Z.G.); qinjianying0203@163.com (J.Q.); 2Longping Branch, Graduate School of Hunan University, Changsha 410125, China; zhouxm1972@126.com

**Keywords:** insect, transcription factors, structures and functions, research methods, progress and prospects

## Abstract

Transcription factors (TFs) play essential roles in the transcriptional regulation of functional genes, and are involved in diverse physiological processes in living organisms. The fruit fly *Drosophila melanogaster*, a simple and easily manipulated organismal model, has been extensively applied to study the biological functions of TFs and their related transcriptional regulation mechanisms. It is noteworthy that with the development of genetic tools such as CRISPR/Cas9 and the next-generation genome sequencing techniques in recent years, identification and dissection the complex genetic regulatory networks of TFs have also made great progress in other insects beyond *Drosophila*. However, unfortunately, there is no comprehensive review that systematically summarizes the structures and biological functions of TFs in both model and non-model insects. Here, we spend extensive effort in collecting vast related studies, and attempt to provide an impartial overview of the progress of the structure and biological functions of current documented TFs in insects, as well as the classical and emerging research methods for studying their regulatory functions. Consequently, considering the importance of versatile TFs in orchestrating diverse insect physiological processes, this review will assist a growing number of entomologists to interrogate this understudied field, and to propel the progress of their contributions to pest control and even human health.

## 1. Introduction

Transcription factors (TFs) are a plethora of proteins that are present in all living organisms, and they can intricately control transcription of different functional genes in response to internal physiological processes, as well as external environmental stimuli [1,2]. Strikingly, it has been estimated that TFs make up approximately 3.5–7.0% of the total number of coding sequences in a reference eukaryotic genome [3,4]. TFs can be divided into diverse classes based on their structural characteristics, and they can influence the transcription of their targets in different mode of actions [5,6]. Most eukaryotic TFs with a DNA-binding domain (DBD) are thought to exert regulatory functions by binding to enhancers and recruiting transcriptional complexes or cofactors [7]. Nevertheless, recent studies have shown that some TFs do not perform the classical functions of binding enhancers and recruiting transcriptional complexes, but rather merely promote regulatory elements such as enhancers and promoters to form looped structures to facilitate transcriptional regulation [8]. TFs have a significant level of functional diversity, and they participate in various biological processes that are pivotal to ensure the proper expression levels of targets in the appropriate time and tissue in organisms.

To date, there are several reviews summarizing the role of certain TF families in certain biochemical processes in insects, especially in *Drosophila* [2,9,10,11,12]. In recent years, the vast range of available genome resources and novel genetic methods greatly promote the identification and investigation of regulatory mechanisms of TFs in diverse insects beyond *Drosophila*, especially lepidopteran insects (Figure 1). However, there is no comprehensive review covering TF structures and functions in different physiological and biochemical processes in *Drosophila* and beyond. Herein, this review describes the structures of the dominating TF types existing in insects, and presents a comprehensive summary of the roles of these TFs in various insect biological processes, which provides new insights into the studies of transcriptional regulation in both model and non-model insects, and elaborates on their application prospects as potential new strategies for pest control in the field, and even as potential targets for human disease treatment and health care.

## 2. Structures

Most TFs contain a DNA-binding domain (DBD) that specifically recognizes and binds to TF-binding sites in the enhancer or promoter of the target gene to read out information and modulate target transcription [13]. Therefore, eukaryotic TFs are generally classified based on their DBD. In humans and mice, TFs are divided into 10 superclasses, the first three of which are the helix-turn-helix (HTH) domain TFs (mostly homeodomain TFs), the basic DBD TFs, and the zinc finger (ZF) TFs [5,14]. These three superclasses occupy 90% and 86% of almost all TFs in humans and mice, respectively [14]. In insects, these three superclass TFs also account for the majority, and we summarize the structures of the three superclasses in insects as follows.

### 2.1. Homeodomain TFs

The HTH superclass TFs contain the HTH motif as the DBD. According to differences in HTH structure, the HTH superclass can be divided into several classes, including homeodomain, Forkhead (Fkh), and Heat shock factor (HSF). Homeodomain proteins play vital biological functions throughout the entire life of insects, especially during growth and development. The homeodomain was originally identified from the homeotic genes in *Drosophila melanogaster* [15,16], which typically contains a short N-terminal arm that contributes to the DNA binding affinity, as well as four α-helices with an HTH structure that is responsible for DNA binding and recognition [16,17]. Helices I and II are antiparallel to each other, helices II and III are separated by a β-turn to form a helix-turn-helix (HTH) structure, and helix III functions as the recognition helix for contacting and recognizing specific DNA sequences (Figure 2A). The homeodomain alone in some homeodomain protein is insufficient for specific DNA binding. Therefore, additional domains and/or cofactors are required to elevate the specificity of DNA binding. For example, Hox proteins generally form homodimers [18] or heterodimers through the conserved YPWM motif upstream homeodomain to increase their selectivity and affinity for DNA binding [19,20]. *Drosophila* contains a total of 103 homeodomain genes, and 54% of TFs are split among families such as paired-like, paired-domain, POU (Pit-Oct-Unc), and LIM proteins that contain other domains for higher binding affinity or protein dimerization, in addition to the homeodomain [21] (Figure 2A).

### 2.2. Basic DBD TFs

#### 2.2.1. bHLH Proteins

The basic helix-loop-helix (bHLH) TFs are widely distributed in insects, and they play crucial roles in the regulation of insect growth and development. The bHLH region contains approximately 60 amino acids, consisting of a basic domain and an HLH domain. The basic domain at the N-terminus is responsible for DNA binding, and it is followed by the HLH region (Figure 2B). In the HLH region, two α-helices that are both hydrophilic and lipophilic are separated by different lengths of a linking loop to form an HLH structure (Figure 2B). In general, bHLH TFs form homo- or heterodimers through α-helix interactions to regulate target transcription [23].

In 2002, Ledent et al. categorized *Drosophila* bHLH proteins and divided them into six groups (A–F) according to their DNA binding and structural characteristics [24] (Appendix A). Group A proteins usually bind the 5′-CAC/GCTG-3′ motif, and they primarily participate in the regulation of mesodermal subdivisions and the development of the nervous system, muscle cells and glands. Group B proteins generally bind the 5′-CACGTG-3′ or 5′-CATGTTG-3′ motifs, and they are involved in the regulation of development, lipid metabolism and glucose tolerance. Group C proteins have a PAS domain followed a bHLH region to combine with a 5′-A/GCGTG-3′ motif [25], which participate in the development of the nervous system and trachea, as well as in the dictation of circadian rhythms and the hypoxia response. Group D proteins contain only an HLH region and they lack a basic domain; these proteins can inhibit the binding of other bHLH proteins to DNA by forming dimers with those bHLH proteins [26]. In *Drosophila*, only one protein Extramacrochaetae (Emc) exists in group D [27,28]. Group E proteins preferentially bind N-boxes (5’-CACGA/CG-3’) and have an additional “Orange” domain and WRPW peptide motif at the C-terminus [29]. Proteins in this group are mainly involved in nervous system development. Group F proteins contain a COE domain, which plays an important role in both dimerization and DNA binding [24,30]. *Drosophila* has only one COE family member, Collier (Col) [31].

#### 2.2.2. bZIP proteins

The basic leucine zipper (bZIP) proteins contain a basic domain that specifically recognizes and binds DNA sequences, followed by a leucine zipper region that is responsible for protein dimerization, and thus promotes the binding of basic regions to DNA [17] (Figure 2B). In the leucine zipper domain, a leucine appears in the seventh position of every seven amino acids, and this region forms dextrorotatory α-helixes, with an adjacent leucine appearing every two turns on the same side of the helix [32] (Figure 2B). Multiple bZIP proteins have been identified in *Drosophila*, and they play essential roles in *Drosophila* development and reproduction (Appendix A).

### 2.3. Zinc-Finger TFs

ZF TFs contain ZF structure motifs as DBDs, and they can be mainly classified into C_2_H_2_, C_4_, and C_6_ classes, according to differences in the conserved ZF domain. TFIIIA is the first ZF protein that has been identified in *Xenopus laevis* in 1983 [33,34]. Subsequently, ZF TFs have been extensively discovered, and they have been found to be the most widely distributed proteins in eukaryotic genomes. In insects, ZF TFs primarily include C_2_H_2_ ZF proteins and nuclear receptors (NRs) with a C_4_ structure (Figure 2C).

#### 2.3.1. C_2_H_2_ Zinc Finger

C_2_H_2_ ZFs can be connected in series to recognize DNA sequences of different lengths [35]. For instance, *Drosophila* Krüppel (Kr) contains four tandemly repeated ZFs [17]. Each C_2_H_2_ ZF is an independent domain with a consensus amino acid motif: X_2_-C-X_2,4_-C-X_12_-H-X_3,4,5_-H (C represents cysteine, H represents histidine, D represents aspartic acid, and X represents any amino acid) [36]. These sequences are tightly folded to form ββα structures in the presence of Zn^2+^, and Zn^2+^ is sandwiched between α-helices and two antiparallel β-sheets [36,37,38]. Two Cys and two His link Zn^2+^ to form a tetrahedron (Figure 2C), in which the two Cys are located on the β-sheet, and the two His are located at the C-terminus of the α-helix. The α-helix acts as a recognition helix to insert into the major groove of DNA and contact specific DNA sequences [17,35].

#### 2.3.2. Nuclear Receptors

NRs in insects are receptors that are responsible for sensing and responding to ecdysone. The C_4_-type ZF domain in NRs is the DBD, and has a consensus sequence: C-X_2_-C-X_13_-C-X_2_-C-X_15,17_-C-X_5_-C-X_9_-C-X_2_-Cys-X_4_-C. This motif consists of eight Cys coordinated with two Zn^2+^ to form two ZFs with a tetrahedral structure (Figure 2C). The first ZF provides DNA binding specificity, and the second ZF has a weak dimerization interface, allowing for the dimerization of the receptor molecule [9]. Besides the conserved DBD, NRs commonly contain a ligand-binding domain in the C-terminus, which is the main region for NR dimerization. NRs typically form dimers to bind DNA motifs, and each receptor molecule recognizes and binds half of the sequence (abbreviated to half-site). The distance between two half-sites and the sequence arrangement facilitate receptor binding to specific DNA motifs [9]. The second ZF plays a critical role in identifying the optimal distance between the two half-sites [17]. Some NRs such as *Drosophila* E75 and E78, can act as “orphan receptors” to bind a single response element.

## 3. Biological Functions

### 3.1. Internal Responses

Numerous TFs in insects precisely regulate spatiotemporal gene expression in response to their internal physiological needs, such as growth and development, metamorphosis, and reproduction. In this regard, the growth and development processes such as embryonic axis establishment, eye development, and gland formation coordinated by substantial numbers of TFs have been thoroughly studied in the model insect *Drosophila*. Additionally, some internal responses of non-model insects modulated by diverse TFs have also recently made great progress. In this section, we systematically review the TF-regulating internal physiological responses in these model and non-model insects.

#### 3.1.1. Embryonic Axis Establishment

In *Drosophila*, the establishment of the embryonic axis has been well documented, and the TF-mediated transcriptional hierarchy has been comprehensively investigated [39,40]. Four morphogens are essential for the establishment of the anterior-posterior (A-P) axis: Bicoid (Bcd) and Hunchback (Hb) regulate the anterior region, and Nanos (Nos) and Caudal (Cad) control the posterior region. Morphogen gradients control the expression of gap genes. Different concentrations of gap proteins activate distinct pair-rule genes, and form seven stripes perpendicular to the A-P axis. Pair-rule proteins in different combinations then activate the transcription of segment polarity genes, further subdividing the embryos into 14 body segments. Finally, Gap, pair-rule, and segment polarity proteins together orchestrate the expression of Hox genes, which determine the developmental fate of each segment (Figure 3).

Dorsal (Dl) protein is critical for the formation of dorsal–ventral (D-V) patterning. The Dl protein is initially uniformly distributed in the egg, but after the ninth cell division, the ventral Dl protein begins to migrate into the nucleus, and the dorsal Dl protein remains in the cytoplasm, causing the formation of a gradient of the Dl protein along the D-V axis. This Dl gradient triggers the formation of mesoderm, neuroectoderm, dorsal ectoderm, and amnioserosa by specifically ordering the expression of different regulatory genes along the D-V axis in early *Drosophila* embryos [43]. A high level of Dl protein activates the expression of *twist* (*twi*) and *snail* (*sna*), and inhibits the expression of *decapentaplegic* (*dpp*) and *zerknüllt* (*zen*) in the mesoderm, and the intermediate levels of Dl activates the expression of *rhomboid* (*rho*), while a low level of Dl activates the expression of *tolloid* (*tld*) and *dpp* in dorsal ectoderm and *zen* in amnioserosa [40,43,44,45,46,47]. Dl-mediated transcriptional repression requires additional factors. Groucho (Gro) is a corepressor of Dl, and it is recruited by Dl to bind to Gro-binding sites closing to Dl-binding sites and other TF-binding sites [46,48,49].

#### 3.1.2. Nervous System Development

*Drosophila* is an excellent model for studying the developmental mechanisms of the central nervous system (CNS). CNS development is a multistep and complex process, and it requires a multitude of TFs to precisely govern the expression of multiple neural development-related genes (Appendix A).

The CNS initially forms in early embryos. In neuroblasts (NBs), TFs are sequentially expressed in a cascade of Hb–Kr–Pdm–Castor (Cas)–Grainyhead (Grh), whose temporal regulation is crucial for the generation of neuronal diversity [50]. Asymmetric cell division (ACD) of embryonic NBs produces two daughter cells: a larger NBs and smaller ganglion mother cells (GMCs). GMCs perform a differentiation function, and they differentiate to produce two neurons or glial cells after mitosis. Snail family proteins play redundant and essential functions in GMC formation by controlling NB ACD [51,52]. Additionally, Worniu (Wor) is continuously required in NBs to maintain NB self-renewal [53].

A much larger and more complicated CNS is established in the larval phase. A different TF cascade (Hth–Ey–Slp–D–Tll) is sequentially activated in the optic lobe NBs, and it regulates temporal expressed genes in type I NBs, resulting in the production of various types of nerve cells [50,54].

Approximately 100 NBs exist in the central brain, the majority of which are type I NBs, whereas only eight are type II NBs. Type I NBs produce terminal dividing GMCs, while ACD of type II NBs first produce immature INPs (imINPs), that need to differentiate into mature INPs before dividing [55]. The marker proteins Asense (Ase) and Prospero (Pros), which determine NB identity, are expressed in type I NBs and mature INPs, but are not expressed in type II NBs. At the end of ACD, Pros is asymmetrically localized to the budding GMC, promoting GMC differentiation.

The ETS family protein Pointed P1 (PntP1) is specifically expressed in type II NBs and imINPs. In type II NBs, Notch signaling inhibits *erm* activation by PntP1, allowing type II NBs to maintain self-renewal and identity [56]. Furthermore, PntP1 represses Ase in type II NBs and promotes the generation of INPs [57]. In imINPs, PntP1 prevents both the premature differentiation and dedifferentiation of imINPs. The Sp family protein Buttonhead (Btd) also functions together with PntP1 to prevent premature differentiation by inhibiting *pros* expression in newly generated imINPs [58].

The ZF protein Earmuff (Erm) is indispensable for INP maturation. In imINPs, *erm* is activated by PntP1 due to a lack of Notch signaling, and by rapid down-regulation of the activities of some self-renewal transcriptional repressors [58,59]. Erm restricts the developmental potential of imINPs, and exerts a negative-feedback effect on PntP1, allowing imINPs to express *ase* and mature [58,60].

#### 3.1.3. Eye Development

The development and formation of *Drosophila* eyes depend on the retinal determination gene network (RDGN), which consists of highly conserved genes encoding TFs and related cofactors that are essential for eye formation [61]. As yet, some TFs are also found to be involved in the regulation of eye development. Orthodenticle (Otd) and ecdysone receptor (EcR) is crucial for the regulation of photoreceptor maturation [62]. Kr regulates the differentiation of photoreceptor neurons (PRs) in the *Drosophila* larval eye [63]. Camta, Lola, Defective proventriculus (Dve) and Hazy are key regulators of PR differentiation in adult ocelli [63].

#### 3.1.4. Trachea and Gland Formation

Gland formation is essential for insect development. The differentiation and morphogenesis of trachea, corpora allata (CA), prothoracic glands (PG), and corpora cardiaca (CC) requires a proper TF cascade during *Drosophila* embryogenesis, respectively [64]. The tracheal epithelial tubes develop from 10 trunk placodes, where Antennapedia (Antp) and STAT activate the expression of *ventral veins lacking* (*vvl*) and *trachealess* (*trh*) for trachea formation. The homologous ectodermal cells in the maxilla and labium form CA and PG, respectively. In the maxilla, Deformed (Dfd) and STAT induce *vvl* and *sna* expression, forming CA with specific-expressed mark *seven-up* (*svp*) after the epithelial-mesenchymal transition (EMT). In the labium, Sex combs reduced (Scr) and STAT regulate *vvl* and *sna* expression, forming PG with the mark gene *spalt* (*sal*) after MET. The CC cells are derived from anterior mesodermal cells, and they specifically express the marker *glass* (*gl*). Likewise, the formation of salivary glands (SGs) in *Drosophila* embryos is also regulated by a series of TFs [65] (Appendix A). SG specification requires Scr and two cofactors, Extradenticle (Exd) and Homothorax (Hth), which work together to activate several early SG TFs, including Fkh, CrebA, Sage, and Huckebein (Hkb). Subsequently, Scr, Extradenticle (Exd), and Homothorax (Hth) disappear, and they are not involved in maintaining SG-specific gene expression. Hkb is transiently expressed in SGs, while Fkh, Sage, and CrebA are continuously expressed in SGs, accounting for the maintenance and implementation of the SG fate decision.

#### 3.1.5. Sex Determination

The sex determination mechanisms exhibit high diversity within and between insects, which promotes the amazing diversity of insects on our planet [66]. The primary signal (e.g., X-chromosome dose, M factor, parental imprinting) that commences sex determination is processed by the master gene (e.g., *sexlethal* (*sxl*), *transformer* (*tra*) or *feminizer* (*fem*)) that carries out alternative splicing and differentially expresses in different genders. The master gene then transmits the sex determination signal to the conserved switch *doublesex* (*dsx*) to control sexual differentiation.

By contrast, the molecular basis of sex determination is well studied in *Drosophila*. *Sxl* is the master factor in *Drosophila* somatic sex determination, which contains two promoters, *Sxl-Pe* and *Sxl-Pm* [67]. In female embryos, the first *Sxl* establishment promoter, *Sxl-Pe*, is transiently activated by a double dose of X-linked signal elements (XSE) or molecular genes to produce the functional protein Sxl, which acts on pre-messenger RNAs (mRNAs) produced by the second constitutive promoter *Sxl-Pm*, to establish the splicing loop and to maintain *Sxl* in an active state [68]. In contrast, in male embryos, *Sxl-Pe* remains inactive, producing a nonfunctional Sxl, and thereby directing the male fate [69]. Several XSE that regulate *Sxl-Pe* have been identified in *Drosophila*, including *scute* (*sc*), *sisA*, *runt* (*run*) and *unpaired* that encode TFs that serve as the primary determinants of X dose [70]. *Sxl-Pm* and *Sxl-Pe* share a common regulatory element (a 1.4 kb region) that responds to the X chromosome dose [67]. Some, but not all, of the X-linked signal TFs that regulate *Sxl-Pe*, including Sc, Daughterless (Da), and Run, are also required for the earliest expression of *Sxl-Pm* [67].

The sex-determining initial signal of other insects is different from *Drosophila*’s X-chromosome dose [71]. In addition, most insects use *tra*/*fem* as the master factor to sense and transmit the primary signal instead of *sxl* [71,72]. Similar to *Drosophila sxl*, *tra*/*fem* autoregulates to produce the corresponding protein and perform gender differentiation.

The conserved TF *dsx* is downstream of master gene, and it is located at the bottom of the sex determination cascade. Most insects contain only one *dsx* gene in their genome, while the generation of multiple *dsx* splice variants (including Dsx^F^ and Dsx^M^) occurs via sex-specific alternative splicing [73] (Appendix A). Dsx isoforms are sex-specifically expressed in the male or female to regulate the expression of gender-related genes which then control sexual differentiation (Appendix A).

The upstream regulation mechanism for the sex determination of the lepidopteran model insect *B. mori* has yet to be fully uncovered, and it is still a research hotspot in developmental biology. Studying and clarifying the sex determination cascade of the representative insect can contribute to our understanding of the insect sex determination molecular mechanism during adaptive evolution, and provide new strategies for pest management.

#### 3.1.6. Wing Imaginal Disc Development

The molecular mechanism underlying insect wing development and differentiation has been a research hotspot for insect development. Decades of research have gradually revealed the molecular mechanism of wing development, especially in *Drosophila*. *Drosophila* wings and legs originate from a common pool of ectodermal cells that express the homeodomain gene *Distal-less* (*Dll*) [74,75]. The concentration of Dpp protein decreases from dorsal to ventral. Under high concentrations of Dpp, the dorsal-most cells expressing *Dll* migrate dorsally and induce the expression of *vestigial* (*vg*) to form the original wing primordium, and later, two ZF genes, *escargot* (*esg*) and *sna*, are expressed to maintain wing disc cell fate [75]. A low concentration of Dpp promotes leg primordium formation from cells expressing *Dll* in the middle and lower regions. After the division and proliferation of wing primordia, four compartments containing different cell populations are generated: cells in the posterior (P) compartment express the homeodomain gene *engrailed* (*en*) while cells in the anterior (A) compartment do not express *en*, and an A-P axis is formed in the first instar larvae; cells in the dorsal (D) compartment express the homeobox gene *apterous* (*ap*), while cells in the ventral (V) compartment do not express *ap*, and a D-V boundary is generated in the third instar larvae [76,77,78]. The proximal–distal (P-D) axis is also required for *Drosophila* wing development. Several TFs such as Stat92E and Zinc finger homeodomain 2 (Zfh2) participate in the establishment of the P-D axis [79,80,81,82]. The wing imaginal disc along the P-D axis is partitioned into the distal pouch, hinge, surrounding pleura, and notum [79]. Hth, Exd, Teashirt (Tsh), and three MADF-BESS family proteins, including Hinge1, Hinge2, and Hinge3, are essential for wing hinge formation [83,84,85].

#### 3.1.7. Lipid Metabolism

Insect fat body is the central organization of insect growth, development, metamorphosis, and reproduction, and many TFs in this tissue play an important role in the regulation of insect lipid metabolism. Among them, FOXO is the major terminal TF for insulin/insulin-like growth factor signaling (IIS), and it modulates lipid metabolism in some insects, including *D. melanogaster*, *Bombyx mori*, and *Glossina morsitans* [86,87,88,89]. Har-Relish responds to 20E signaling, and it regulates fat body dissociation in *Helicoverpa armigera* [90]. In *Drosophila*, activating transcription factor-2 (ATF-2) and βFTZ-F1 participate in lipid metabolism [91,92]. In *Aedes aegypti*, hormone receptor 3 (HR3), Thanatos-associated protein (THAP) and ATF-2 regulate the transcription of *Sterol carrier protein 2* (*SCP2*), which is a critical factor for sterol absorption and transport [93,94]. Moreover, C/EBP may directly regulate *SlSCPx* expression in *Spodoptera litura* [95].

#### 3.1.8. Circadian Clock Adjustment

The circadian clock system of most living organisms participates in the regulation of various rhythmic behaviors and physiological functions. The *Drosophila* circadian rhythm is mainly regulated by a transcriptional translation feedback loop (TTFL) that is centered on the master circadian transcription complex Clock-Cycle (CLK-CYC). The CLK-CYC heterodimer directly activates the transcription of the core circadian genes *period* (*per*) and *timeless* (*tim*) during the night [96]. Phosphorylated Per and Tim in turn repress the transcription of *clk* and *cyc* [97]. Later, Per and Tim are degraded in the presence of light, allowing CLK-CYC to initiate the next cycle of transcription [98]. In addition, CLK-CYC activates the expression of hundreds of clock-controlled output genes such as *vrille* (*vri*), *Par Domain Protein 1ε* (*Pdp1ε*), *clockwork orange* (*cwo*), and *Mef2* [99,100,101]. In turn, VRI, PDP1m and Cwo can also regulate the expression of *clk* and *cyc* [99], and Mef2 is involved in neuronal remodeling to facilitate locomotor activity rhythms [101]. Meanwhile, the NRs induced by ecdysone are also involved in the regulation of insect rhythms. E75 and Unfulfilled (UNF; DHR51) strengthen the CLK/CYC-mediated transcription of *per* by directly binding to the regulatory element [102]. In firebrat *Thermobia domestica*, the normal circadian expression of *E75* and *HR3* is necessary for the maintenance of locomotor rhythms [103].

#### 3.1.9. Diapause Control

Diapause can facilitate insect survival from adverse environmental conditions, such as extreme weather or reduced food availability. In *H. armigera*, multiple TFs, including Har-AP-4, POU-M2 (Vvl), and FoxA control diapause by directly regulating the expression of *diapause hormone* and *pheromone biosynthesis-activating neuropeptide* (*DH-PBAN*) [104,105,106,107]. In *B. mori*, POU-M2 is also essential for the regulation of *DH-PBAN* [108], and BmILF is involved in the transcriptional regulation of *POUM2* [109]. Additionally, the diapause status of *Pyrrhocoris apterus* guts is triggered under a short photoperiod in winter. Low-level JH leads to *cry2* expression overriding *Pdp1_iso1_*, thus initiating the diapause-specific program and activating the expression of the diapause downstream genes *superoxide dismutase* (*sod*) and *transferrin* (*tf*) in the gut [110].

#### 3.1.10. Cuticular Protein Synthesis

Insect cuticular protein is one of the main components that constitute the insect stratum corneum. Spatiotemporal expression of insect cuticular protein genes (ICPGs) is regulated by multiple TFs, especially a series of 20E-response genes (Appendix A). For example, the NR βFTZ-F1 is one of the early found TFs to regulate IGPG expression, it can regulate the expression of *EDG84A* and *EDG74E* in *Drosophila* [111], MSCP14.6 in *Manduca sexta* [112], and many ICPGs in *B. mori*. In addition, the homeoprotein BmPOUM2 interacts with BmAbd-A to regulate *BmWCP4* gene expression [113,114]. Although several TFs that regulate some ICPG expression have been identified, multiple uncharacterized TFs that control other ICPGs warrant further investigation.

#### 3.1.11. Cuticle Coloration Dictation

Insect body color and markings pattern are of great significance for insect survival and reproduction. Abundant coloring patterns are displayed in butterfly adult wings and in the epidermis of silkworm larvae and different species of fruit fly. Pigment patterning in *Drosophila* adults has been intensively studied, and is regulated by pleiotropic regulatory TFs, including sex-determination genes (e.g., *Dsx*), HOX genes (e.g., *Abdominal-B* (*Abd-B*)), and selector genes (e.g., *Optomotor-blind* (*Omb*) and *Engrailed* (*En*) via the control of the expression of effector genes that encode the enzymes and co-factors required for pigment biosynthesis [115,116]. The butterfly wing pattern is also regulated by multiple patterning TFs such as Omb, Abd-B, Dsx, Sal, and En [116,117]. The TF Apt-like participates in *B. mori* larval epidermal pigmentation or the melanin biosynthesis pathway by regulating the single genetic *p* locus that contains at least 15 alleles, and produces a phenotypic diversity of pigment patterns [118].

Insect hyperpigmentation is a good model for studying insect adaptation, evolution and development. Pigmentation diversity in insects can be attributed to changes in the expression levels of transcriptional activators and changes in the *cis*-regulatory elements of the pigment synthesis gene for TF binding [115]. Deciphering the mechanism of insect coloration regulated by TFs provides an important reference for agricultural and forestry pest control, and ecological adaptability exploration. Nonetheless, the current accumulated knowledge is not enough to allow us to fully understand the complete regulation network of insect coloring pattern; thus, further studies are required to identify regulatory TFs and to expound the regulatory mechanism.

#### 3.1.12. Silk Protein Production

Silk is mainly composed of fibroin and sericin. Fibroin consists of the fibroin heavy chain (fibH), fibroin light chain (fibL) and P25 proteins. These genes are specifically expressed in posterior silk gland (PSG) cells during the feeding stage of silkworm larval development, but they are suppressed during the molting stage. The *sericin-1* (*ser1*) gene is expressed in the posterior of the middle silk gland (MSG) before the fifth instar larvae, and its expressional region extends to the middle in the fifth instar larvae. A variety of TFs jointly regulate the spatiotemporal expression of these silk protein synthesis-related genes (Appendix A). Many TFs have been reported to regulate *fibH* gene expression. Among them, the bHLH TFs Dimmed (Dimm) and Sage usually form heterodimers with other proteins to regulate *fibH* expression. For instance, Dimm directly activates *fibH* expression by interacting with Sage [119]. Dimm can also act as a repressor of *fibH* by interacting with repressor MBF2 [120]. Sage forms a complex with Fkh to enhance *fibH* expression [121]. Whether Dimm, Sage, and Fkh can form a triplet to activate *fibH* transcription merits further study [119]. Relatively few TFs are known to regulate *fibL* and *P25* genes. Fkh and SGF2 positively regulate the expression of the *fibL* gene [122,123]. Fkh, SGF2, PSGF, and BMFA are involved in the regulation of *P25* expression [124,125,126]. Some TFs that positively regulate the expression of the *sericin-1* gene have also been identified, including Fkh [127], POU-M1 [128], and Antp [129]. Additionally, POU-M1 participates in the restriction of the anterior boundary of the *ser1* expression region [130]. Nevertheless, although many transcriptional activators controlling the expression of silk protein synthesis-related genes have been identified, transcriptional repressors inhibiting the expression of these genes at the molting stage, thereby limiting their spatial expression, still remain largely unknown.

#### 3.1.13. Molting and Metamorphosis Initiation

Insect larval molting and metamorphosis are coordinated by ecdysone and juvenile hormones (JHs). The 20-hydroxyecdysone (20E, the biologically active form of ecdysone) induces larval–larval molting in the presence of JHs, while 20E induces larval–pupal and pupal–adult metamorphosis upon the disappearance of JHs [131].

20E regulates various physiological and biochemical processes in insects, especially molting and metamorphosis [132]. TFs play an essential role in the regulation of ecdysone titers. Several TFs have been identified to specifically regulate Halloween genes encoding a series of ecdysone biosynthetic enzymes, to promote steroidogenesis (Figure 4A and Appendix A). Among them, Séance (Séan), Ouija board (Ouib), and Molting defective (Mld) are only found in *Drosophila*, they are, therefore, thought to be evolved specifically to control the transcription of the two Halloween genes *neverland* (*nvd*) and *spookier* (*spok*) in *Drosophila* [133]. Reduction of ecdysone titers regulated by TFs occurs in two ways: inhibition of Halloween gene expression and the direct degradation of ecdysone (Figure 4A). Hence, changes in ecdysone titer in insects are regulated by TFs via manipulating the synthesis and degradation of ecdysone. Accordingly, the 20E regulatory cascades have been proposed [9]. In general, 20E signaling is transduced by NRs. Firstly, 20E binds to the EcR/Ultraspiracle (USP) complex, and then the 20E/EcR/USP complex directly induces the early 20E-response genes including *E74, E75*, and *Broad-Complex* (*Br-C*). Products of these early genes activate the later 20E-response genes, which encode TFs to regulate the spatiotemporal expression of downstream targets. Furthermore, the expression of some of the 20E-response genes is also controlled by both 20E/EcR/USP and early responsive products.

JHs are synthesized and secreted by the corpora allata (CA) in insects. The prominent role of JHs is to prevent the premature transition of immature larvae to pupae and adults [131]. There have been several studies on JH regulation by TFs, as summarized above (Figure 4B). In *Drosophila* CA cells, phosphorylated Mad shuttles into the nucleus, together with co-Smad, and triggers the expression of the JH biosynthetic enzyme, *jhamt* [134]. TcVvl is upstream of JH signaling, and it is important for the normal expression of the JH synthetic gene *jhamt3* [135]. In addition, BmFOXO regulates JH degradation by regulating the expression of JH-degrading enzyme genes *JHE*, *JHEH*, and *JHDK* [136]. In insects, JH signaling is primarily transduced by the JH receptor Met, which is a member of the bHLH-PAS family and was originally identified in *Drosophila* mutants [137]. Met typically forms dimers to directly regulate target gene transcription. SRC (also known as “FISC” or “Taiman”, hereinafter referred to as SRC) is the most common coactivator of Met in multiple insects, including *Drosophila, A. aegypti*, *B. mori*, and *Tribolium castaneum* [138,139,140,141,142,143]. In *Drosophila*, Met also forms homodimers or forms heterodimers with another bHLH-PAS family member, Gce, to perform functions [144,145]. The JH/Met-SRC/Krüppel homolog 1 (Kr-h1) cascade is conserved in both holometabolous and hemimetabolous insects, and it mediates JH-repressed metamorphosis. The *Kr-h1* gene is a direct target of Met, and it encodes a C_2_H_2_ ZF protein that plays a central role in the JH-mediated inhibition of metamorphosis [142,143,146,147,148]. Kr-h1 inhibits the expression of the pupal specifier *Br-C* to prevent premature metamorphosis from larva to pupa in the larval stage [134,149]. The transient peak of *Kr-h1* at the end stage of the last-instar larvae upregulates the expression of *Br-C* to allow for the correct formation of the pupae, and inhibits the premature upregulation of *E93* to prevent larvae from bypassing the pupal stage and directly developing into adults [150,151].

#### 3.1.14. Reproduction Manipulation

One of the main reasons of pest outbreaks is the high fecundity of insects based on oogenesis, which can be divided into three developmental stages: previtellogenesis, vitellogenesis, and choriogenesis. Insect oogenesis is a complicated biological process that is coordinatively controlled by various signaling pathways, especially the 20E and JH signaling pathways, as well as multifarious TFs.

JH-regulated insect reproduction is mediated through the JH-receptor complex Met–SRC. In the previtellogenesis of *A. aegypti*, Met–SRC regulates the expression of downstream genes, preparing for subsequent vitellogenesis and egg development [152]. Met is capable of directly activating target transcription, while the suppressing action of Met on targets is indirect and requires other mediators such as Hairy [153]. Studies have shown that Hairy and its corepressor Gro in female *A. aegypti* mediate the repression of 15% of Met-repressed genes [154]. Further studies are required to reveal other mechanisms mediated Met action in gene repression [155]. In the migratory locust, the Met–SRC complex directly regulates the expression of *Mcm4*, *Mcm7*, and *Cdc6* to promote DNA replication and polyploidy for vitellogenesis and oocyte maturation [156,157]. The complex also induces the expression of Grp78-2, which is required for insect fat body cell homeostasis and vitellogenesis [158]. In addition, Met-SRC directly activates the transcription of *Kr-h1* to promote vitellogenesis and oocyte maturation [159]. In *Cimex lectularius*, Met–SRC also regulates vitellogenesis and ovigenesis by the indirect regulation of *Vg* synthesis, but TFs downstream of Met that regulate the expression of *vitellogenin* (*Vg*) still remain mysterious [160]. In female *P. apterus*, the Met–SRC complex is required for JH-induced *Vg* expression during vitellogenesis [148]. Regulation of the reproductive status of the *P. apterus* gut requires Met, as well as its cofactors CLK and CYC, to activate the expression of *Pdp1_iso1_*, which in turn upregulates the reproduction-associated genes *lipase* (*lip*), *esterase* (*est*) and *defensin* (*def*), and suppresses *Cryptochrome 2* (*Cry2*) and the diapause-related downstream genes *superoxide dismutase* (*sod*) and *transferrin* (*tf*) [110]. Met–SRC is also involved in the regulation of male reproduction by controlling the accessory gland proteins and hexamerins in fat bodies of male *P. apterus* [161].

Ecdysteroid-dependent regulation of insect oogenesis is induced by a series of NRs and 20E response genes [152]. In *A. aegypti*, 20E regulates the vitellogenesis of female mosquitoes by regulating the transcription of the *Vg* gene. In addition to regulation of midoogenesis, oocyte maturation and oviposition, ecdysone regulates the very early steps of oogenesis, including niche formation, germline stem cell (GSC) behavior, and cyst cell differentiation in *Drosophila* [152,162].

Most studies on the regulation of chorion gene expression by TFs mainly focus on the insect model *B. mori* during the choriogenesis period [2]. The CCAAT-enhancer-binding protein (C/EBP) is a major regulator of early/early-middle chorion gene expression in *B. mori* [95]. The relative concentration of C/EBP is correlated with its differential binding affinity to the response elements, leading to the activation or repression of targets [163]. Another two C/EBP-like proteins, the chorion bZIP factor (CbZ) and C/EBPγ, can form the CbZ-C/EBPγ heterodimer to repress chorion gene expression by antagonizing the binding of C/EBP homodimers to the promoter [164]. The expression of the late chorionic gene in silkworm is generally regulated by *Bombyx* Chorion Factors I (BCFI) and GATAβ. The Forkhead box transcription factor L2 (NlFoxL2) in *Nilaparvata lugens* directly activates *follicle cell protein 3C* (*NlFcp3C*) to regulate chorion formation [165].

### 3.2. External Responses

Insect TFs also play an important role in the external response, which can improve the tolerance of insects to adverse environments and protect them from diverse external stress.

#### 3.2.1. Biotic Factor Responses

Insect responses to biotic factors primarily includes immune responses to pathogens and viruses. TFs play an essential role in humoral immunity, especially in the regulation of the production of antimicrobial peptides (AMPs).

In insects, the regulation of inducible AMP genes relies mainly on NF-κB factors that are activated by the intracellular Toll or Imd signaling pathway when infected by pathogens and parasites. *Drosophila* is used as a model to investigate the innate immune mechanism. Three NF-κB proteins have been identified in *Drosophila*: Dl, Dorsal-related immunity factor (Dif), and Relish. These proteins contain the N-terminal Rel-homology domain (RHD) that is used for DNA binding and dimerization [166]. However, only Relish contains an inhibitor of inhibitor κB (IκB) domain [167]. These NF-κB proteins are activated by two distinct pathways: the Toll and Imd signaling pathways. When *Drosophila* are infected with Gram-positive bacteria, fungi, and viruses, the Toll pathway is responsible for the activation of the NF-κB proteins Dl and Dif. It first causes the dissociation of Dif–Cactus and Dorsal–Cactus complexes in the cytoplasm, and then Dif and Dl translocate to the nucleus and activate the expression of specific AMP genes such as *Drosomycin* (*Drs*) [168,169,170]. The immune deficiency (Imd) pathway is activated upon infection by Gram-negative bacteria, leading to the endoproteolytic cleavage of the Relish protein in the cytoplasm. Subsequently, its N-terminal fragment containing RHD translocates to the nucleus and activates expression of AMP genes [171].

NF-κB factors can form homodimers or heterodimers to regulate AMPs expression. The *Drosophila* Dif-Relish heterodimer linked by a flexible peptide linker can activate *Diptericin* (*Dipt*) and *CecA1* [172]. However, it is still unclear whether Dif–Relish or Dl–Relish heterodimers actually form *in vivo* [172,173]. In addition, several cofactors interacting with NF-κB proteins have also been identified in *Drosophila*, including the coactivators Dip4/Ubc9 [174] and Dip3 [174,175], the three POU proteins Pdm1, Pdm2, and Dfr/Vv1 [176], as well as corepressors such as Cautus and Gro [46,48,168].

The homologs of *Drosophila Dl* and *Relish* have been found in other insects, which also regulate the expression of inducible AMP genes in responses to pathogens and parasites (Appendix A). *A. gambiae* and *A. aegypti* have two NF-κB genes: *REL1* and *REL2*, which are the homologues of *DmDl* and *DmRelish*, respectively [177,178,179,180,181,182]. *AgREL2* gene encodes two *REL2* isoforms REL2-Full and REL2-Short, through alternative splicing of the *REL2* gene [179]. The IMD/REL2-Full cascade defends against Gram-positive *Staphylococcus aureus*, and regulates the intensity of mosquito infection with the malaria parasite *Plasmodium berghei*, whereas REL2-Short is resistant against the Gram-negative *Escherichia coli* [179]. The *AaREL1* gene encodes two isoforms, AaREL1-A and AaREL1-B, which are the key activators of the Toll-mediated antifungal immune pathway to activate the expression of *Dipt* and *Drs*, and elevate defense against the fungus *Beauveria bassiana* [180]. The *AaREL2* gene encodes three isoforms: REL2-Full, containing the RHD and the IkB-like domain, REL2-Short, comprising RHD, and REL-IkB, with only the IkB-like domain [183]. All three *Relish* transcripts are activated when *A. aegypti* is infected by bacteria [183]. REL2 prevents *Aedes* against infection by Gram-positive and Gram-negative bacteria and *Plasmodium gallinaceum* [184,185]. In *Culex quinquefasciatus*, the *DmRelish* homolog *Rel2* is activated by a TEAE-dependent pathway after WNV infection, and binds to the NF-κB site of the upstream promoter of the *Vago* gene to induce *Vago* expression, thereby triggering antiviral responses [186]. In *B. mori*, the *BmRel* gene, a homolog of *Dl*, encodes two isoforms: BmRelA (long) and BmRelB (short). These two isoforms act differentially to activate antibacterial peptide genes: BmRelB strongly activates the *Attacin* (*Att*) gene, while *BmRelA* strongly activates the *lebocin 4* (*Leb4*) gene and weakly activates the *Att* and *lebocin3* (*Leb3*) genes [187]. *BmRelish* (gene homologous to *Drosophila Relish*) also encodes two proteins: BmRelish1 and BmRe0lish2. BmRelish1 can activate the expression of *CecB1*, *Att* and *Leb4*. BmRelish2 is a dominant negative factor of the *BmRelish1* active form, and it inhibits the *CecB1* gene activated by BmRelish1 [188]. There are also two NF-κB genes in *M. sexta*: one is *MsDorsal* (the homologous gene of *DmDl),* and the other is *MsRel2* (the homologous gene of *DmRelish*), which produces two isoforms, MsRel2A and MsRel2B [173]. These three NF-κB factors can form homodimers and activate promoters of different AMP genes. Moreover, *MsDorsal* and *MsRel2* form heterodimers to repress the activation of AMP gene promoters and prevent their overactivation [173].

In addition to NF-κB proteins, other TFs are also involved in the regulation of inducible AMP expression (Appendix A). However, the relationship between NF-κB proteins and Toll and Imd pathways remains to be clarified in other insects beyond *Drosophila*.

Constitutive AMP genes are expressed continuously in an NF-κB-independent manner in defined tissues, to function as a first line of defense against microbial infection during development and reproduction. For example, the *Drosophila* homeodomain protein Cad regulates the continuous expression of *Cec* and *Drs* in epithelia in a NF-κB-independent manner [189]. Cad also regulates commensal–gut mutualism by inhibiting NF-κB-dependent AMPs [190]. The POU protein Vvl synergizes with other proteins to regulate constitutive AMP gene expression in a range of immunocompetent tissues, including the male ejaculatory duct [191]. In *M. sexta*, Fkh activates a series of AMPs under non-infectious conditions to protect them from microbial infections during insect molting and metamorphosis [192]. This activation is possibly essential for the defense against microbial infection during insect molting and metamorphosis [192]. During *B. mori* metamorphosis, 20E activates Br-Z4 and Ets to regulate *Leb* expression in the midgut to protect the midgut from infection [193].

Insect immune response is highly homologous to mammals, and insects are relatively simple and easy to manipulate, compared with mammals. Therefore, the study of insect TF regulating immune response can not only enable us to understand the entire immune system in insects, but also inspire our understanding and exploration of the human immune regulatory mechanism. Thus far, the studies on AMP gene expression regulated by insect TFs have focused on holometabolous insects such as fruit fly, mosquito, silkworm, etc., while there are few studies on hemimetabolous insects. It is possible that the TF immunoregulatory mechanisms in hemimetamorphosis insects is different from holometabolous insects, because functional annotation of immune and defense-related genes in the aphid genome revealed that some of the AMPs commonly found in metamorphosis insects are not expressed in aphids [194]. Therefore, studies of TFs regulating immune responses in insects can be concentrated in hemimetamorphosis insects in the future.

#### 3.2.2. Abiotic Factor Responses

Insect abiotic factor responses are mostly comprised of the resistance to xenobiotics (including chemical pesticides, biological pesticides, and secondary metabolites), and the response to high temperature and oxygen stress. Numerous studies have shown that diverse TFs are involved in xenobiotic resistance in insects (Appendix A).

CncC and aryl hydrocarbon receptor (AhR) can regulate insect tolerance to plant secondary toxicants. CncC participates in the *Leptinotarsa decemlineata* adaptation to potato plant allelochemicals and *Aphis gossypii* tolerance to gossypol [195,196]. AhR heterodimerizes with aryl hydrocarbon receptor nuclear translocator (ARNT) to directly activate the expression of P450s after exposure to plant secondary metabolites [197,198]. In addition, two other potential TFs (FK506 binding protein (FKBP) and Prey2) were reported to regulate the expression of *CYP6B6* in *H. armigera* under 2-tridecanone stress [199].

Insect resistance to chemical insecticides is regulated by TFs such as CncC and DHR96 [200,201]. CncC extensively participates in insect resistance to different insecticides other than plant secondary toxicant [200]. Although CncC has a short half-life, its constitutive activation in some insects can confer a resistance phenotype [200]. Recently, a genome-wide analysis of TFs in *Plutella xylostella* found that the altered expression of multiple TFs may be involved in *P. xylostella* insecticide resistance, but their precise functions remain to be further validated [202]. Moreover, it has been reported that some unidentified TFs downstream of the G protein-coupled receptor (GPCR) signaling can be involved in regulation of P450-mediated permethrin resistance in *C. quinquefasciatus* [203,204].

Insect resistance to biopesticides such as *Bacillus thuringiensis* (Bt) can also be modulated by some TFs. Altered expression of midgut functional genes can lead to Bt resistance in many insects, but the potential TF-mediated regulation mechanisms of their expression alteration still remain to be unveiled [205]. For example, our previous studies have shown that high-level resistance to Bt Cry1Ac toxin in *P. xylostella* is associated with differential expression of a suite of midgut functional genes, including *ALP*, *ABCC1*, *ABCC2*, *ABCC3*, and *ABCG1*, which are *trans*-regulated by the MAPK signaling pathway [206], and we can speculate that the novel MAPK-mediated *trans*-regulatory mechanism may be further controlled by diverse downstream TFs such as FOXA [207]. Further study deserves to be conducted in order to characterize these downstream TFs, to comprehensively understand Bt resistance mechanisms in different insects.

Under heat shock or other stresses, HSF relocalizes within the nucleus to form a trimer to activate heat shock (HS) gene expression by binding to HS elements [208,209]. In addition to *HS*, some genes under non-stress conditions might be the targets of HSF, since that HSF is also required for oogenesis and early larval development under normal growth conditions [210].

The HIF family member Similar (Sima) is required for the hypoxia response and normal development in *Drosophila*. Under hypoxic conditions, Sima is upregulated and heterodimerizes with Tgo to activate the expression of related genes [28]. In addition, Sima activates Notch signaling to facilitate the survival of *Drosophila* blood cells under both normal hematopoiesis and hypoxic stress [211].

Altogether, countering the selection pressures of non-biological factors offers an evolutionary force for insects to adapt to the surrounding environment, and TFs play a pivotal role in this adaptive evolution process. Unfortunately, little is known about the TFs that are response to resistance-related signaling cascades and that regulate the expression of xenobiotic-resistant genes. Thus, this area will become a research hotspot, and it will facilitate insect resistance management in the near future.

## 4. Research Methods

### 4.1. TF-Binding Site (TFBS) Prediction

TFs regulate the transcription of target genes by specifically binding to their TFBS located in the regulatory region. Therefore, TFBS prediction in the target promoter is a critical step to studying transcriptional regulation. A number of databases specifically collecting TFBS-related information have been established, with the advancement of experimental techniques for TFBS identification, and multiple online software and websites have been developed with the rapid development of bioinformatics, which allows researchers to predict TFBS in target promoters *in silico*, which lays a critical foundation for further transcriptional regulation studies (Table 1).

### 4.2. DNA–Protein Interaction Detection

TFs act mainly through binding directly to sequence-specific DNA motifs in the promoters or enhancers of target genes. Therefore, identifying the interaction between TFs and DNA is particularly crucial for TF functional studies. In this section, we comprehensively elaborate the basic principles, merits, faults, and applications of several current techniques that are extensively applied in investigating DNA–protein interactions, which will provide theoretical and technical guidelines for researchers to study TF functions.

#### 4.2.1. Dual-Luciferase Reporter Assay System

The luciferase reporter gene assay is extensively used for detecting the interaction between TFs and DNA motifs, and it is characterized by a high level of sensitivity, good specificity, short detection time, and a wide linear range. Researchers often use one luciferase (such as firefly luciferase) to monitor gene expression, and another type of luciferase (such as *Renilla* luciferase) as an internal control to construct the dual-luciferase reporter assay system that reduces external interference and improves system detection sensitivity and reliability [220]. Since the introduction of the dual-luciferase assay system in the mid-1990s, this reporter assay has been widely applied in the study of TF–DNA interaction [221,222,223]. For example, the functional interaction between Kr-h1 and Kr-h1 binding site (KBS) in the E93 promoter was examined by the dual-luciferase assay system in *B. mori* [131].

#### 4.2.2. Electrophoretic Mobility Shift Assay (EMSA)

EMSA is a classical technique for the rapid and sensitive investigation of protein–DNA interactions *in vitro*, it has the advantages of simple operation and high detection sensitivity. The currently widely used assays are based on methods originally described by Garner and Revzin [224] and Fried and Crothers [225,226]. The labeled nucleic acid probes bind protein to form nucleic acid–protein complexes that migrate more slowly in gel electrophoresis than do the corresponding free nucleic acids, whereupon the nucleic acid–protein complexes are separated out. Additionally, it can also use competitive experiments and supershift assays to evaluate the properties of protein–DNA binding. For instance, EMSA was succeeded to confirm the specific binding of BmE74A to the E74A binding site in the *ecdysone oxidase* (*EO*) promoter in *B. mori* [227]. However, this method *in vitro* does not truly reflect the interaction between proteins and nucleic acids in organisms [228]. Moreover, this method requires nucleic acids be labeled with radioisotopes, fluorophores, or biotin, which takes a long time and has a high cost. In recent years, emerging microfluidic-based EMSAs were not just limited to the investigation of protein–nucleic acid interactions; these assays include high-throughput and multiplexed analyses that could be applied for molecular conformational analysis, immunoassays, affinity analysis and genomics study [229].

#### 4.2.3. Yeast One-Hybrid System (Y1H)

Y1H assay is an effective method for studying protein–DNA interaction in yeast cells. The Y1H assay consists essentially of two components: a reporter gene that contains the known specific DNA sequence, and a construct that contains the complementary DNA (cDNA) encoding the test TF that is fused to the activation domain (AD) of the yeast Gal4 (Gal4-AD), both of which are transferred to a suitable yeast strain [230]. If the TF can bind to the DNA sequence, Gal4-AD will activate reporter gene expression [231]. Of course, the Y1H assay also has some disadvantages. It usually takes a long time and has difficultly detecting interacting dimers or proteins that require posttranslational modifications to bind DNA [230]. In addition, it may lead to some uncharacterized TFs binding to the target DNA, owing to the incompleteness of the TF library in a species.

#### 4.2.4. Chromatin Immunoprecipitation (ChIP)

ChIP is based on the principle of antigen–antibody binding, and it can reflect the interactions of proteins and DNA that occur in living cells. ChIP was originally used for the study of histone covalent modification, and was later widely used in the study of TF–DNA interaction. ChIP generally first fixes the protein–DNA complexes that occur in the cell with formaldehyde. The cells are then lysed, and the chromatin is randomly cleaved into small segments of a certain length. The protein–DNA complexes are then selectively immunoprecipitated using specific protein antibodies against the target protein. DNA fragments that bind to the target protein are then specifically enriched, purified, and identified [232,233].

There are three predominant methods for the identification of immunoprecipitated DNA: ChIP-qPCR, ChIP-chip and ChIP-seq. ChIP-qPCR is the earliest method to identify the specific binding of proteins to DNA and it is suitable for identifying the known sequence of precipitated DNA fragments and quantifying the binding of TFs to specific target DNA [233]. ChIP-chip has become a common method for studying protein–DNA interactions since Ren et al. applied the ChIP-chip method for the first time to identify the genome-wide binding sites of the transcriptional activators Gal4 and Stel2 in yeast [234]. The ChIP-seq method was first reported in 2007, which combines ChIP with massively parallel DNA sequencing, and can efficiently detect genome-wide DNA fragments that interact with TFs or histones [235]. ChIP-chip and ChIP-seq do not require knowledge of the target DNA sequences in advance, and it can identify whole-genome targets and quantify binding levels [233]. ChIP-seq can quickly decode a large number of DNA fragments at a higher efficiency and at a relatively low cost, compared with ChIP-chip [232]. In addition, the data provided by ChIP-seq are of higher resolution, and the information obtained is more accurate and quantitative than that from ChIP-chip [232]. Thus, ChIP-seq is currently one of the most frequently used methods for studying protein–DNA interactions. Recently, the genome-wide ChIP-seq analysis in *B. mori* has identified a consensus KBS in the *E93* promoter [131], which provides a paradigm to use this technique in insects.

ChIP has a broad application prospect and can capture the interaction between TFs and binding sites *in vivo*, and identify the distinct regulatory mechanisms of differentially expressed genes [232]. However, this approach has its limitations as well: it requires highly specific antibodies. Acquisition of highly abundant binding fragments requires a high level of simulation of an intracellular environment that is required for target protein expression. In addition, it is difficult to simultaneously obtain information on the binding of multiple proteins to the same sequence [236].

#### 4.2.5. CRISPR Affinity Purification In Situ of Regulatory Elements (CAPTURE)

Recently, researchers have developed a new approach, CAPTURE, to isolate chromatin interactions in situ by using the targeting ability of the CRISPR/Cas9 system and high affinity between biotin and streptavidin [237]. CAPTURE includes three key components: an N-terminal FLAG and a biotin-acceptor site (FB)-tagged deactivated Cas9 (dCas9), a biotin ligase (BirA), and a single guide RNA (sgRNA) that serves to direct biotinylated dCas9 to the target genomic sequence. Upon *in vivo* biotinylation of dCas9 by the biotin ligase BirA together with sequence-specific sgRNAs in mammalian cells, the genomic locus-associated macromolecules are isolated by high-affinity streptavidin purification. The purified protein, RNA, and DNA complexes are then identified and analyzed by mass spectrometry (MS)-based proteomics and high-throughput sequencing for the study of native CRE-regulating proteins, RNA, and long-range DNA interactions, respectively. This approach is more specific and sensitive than ChIP, and it does not require protein antibodies and the known TFs. Considering these advantages of CAPTURE, we believe that this method will also be applied for *in vivo* TF–DNA interaction detection in insects in the near future.

### 4.3. TF Function Verification

#### 4.3.1. CRISPR/Cas9 system

The novel CRISPR/Cas9 technology has been widely used to modify genome sequences in multiple species recently [238]. At present, researchers have begun to apply the CRISPR/Cas9-mediated genome editing system to investigate the regulatory function of TFs in organisms, including insects, plants, and crustaceans [136,239]. The study of TF regulatory function by CRISPR/Cas9 can be divided into two categories. One is to mutagenize the exon of the TF locus through CRISPR/Cas9, and to generate mutants to study TF functions in insects, including *Drosophila*, *B. mori*, and *P. xylostella* [136,239,240], and the other is to knockout the TF-binding site on the promoter of target gene, and then to observe the transcription level of the target gene to study the function of the TF in the crustacean *Daphnia magna* [241].

#### 4.3.2. Yeast Two-Hybrid Assay (Y2H)

TFs often function as homodimers or heterodimers. Hence, understanding the mechanisms of protein–protein interactions is essential for determining the actions of TFs. The Y2H assay has been widely used to identify protein–protein interactions since its appearance in 1989 [242]. As for insects, for example, Bric-a-brac interacting protein 2 (BIP2) has been confirmed to be an ANTP-interacting protein by using this assay in adult *Drosophila* [20]. In the Y2H assay, two proteins are fused into the DBD and AD of Yeast Gal4, respectively. If these two proteins interact with each other, an active Gal4 TF would be generated and induce the transcription of *lacZ* reporter gene in yeast cells. The initial Y2H had some limitations, such as not reflecting complex spatial or temporal interactions *in vivo*. The continuous improvement of Y2H technique has not only overcome the major limitations of the original Y2H system, but also has expanded its application areas. In particular, the development of high-throughput Y2H allows it to be applied for the investigation of complex protein interactions [243,244,245]. Furthermore, Y2H has also been used to study other types of molecular interactions and to identify domains that stabilize protein–protein interactions [246,247].

#### 4.3.3. Expression Read Depth GWAS (eRD-GWAS)

Many phenotypic changes in organisms are caused by changes in the expression patterns of various regulatory genes, such as genes encoding TFs. eRD-GWAS is a genome-wide association studies based on Bayesian analysis using gene expression level data tested by RNA-Seq as an explanatory variable. This method can identify true relationships between gene expression variation and phenotypic diversity at the genomic level, and it is an effective complement to SNP-based GWAS. Lin et al. applied this method in maize, and revealed that genetic variation in TF expression contributes substantially to phenotypic diversity [248]. Apparently, we can anticipate that this novel and promising method will also be adopted to validate the TF functions in insects.

## 5. Discussion and Prospects

Evidently, TFs play a central role in the insect genetic regulatory network, as in other organisms. In this review, we integrate vast amounts of TF information in both model and non-model insects, and summarize their vital functions in response to internal signaling and external stimuli.

Probing TFs in the model insect *D. melanogaster* has yielded fruitful results, which provide important insights into the study of TFs in other organisms. In recent years, TF studies in other insects have also achieved great success in the development of genetic tools and next-generation genome sequencing techniques. However, there are still large numbers of unidentified TFs and uncharacterized TF functions in insects. Moreover, understanding the precise regulatory mechanism of TFs still remains a great challenge [249]. Although TF–TF interactions and TF–DNA interactions are prevalent in organisms, both of these interactions are largely undetectable because they depend not only on the opening degree of chromatin, but also on whether the interaction is instantaneous or long-term, and how strong the interactions are. More research is required to understand how TFs interact with specific DNA sites to regulate the spatiotemporal expression of target genes, and how TFs interplay to achieve regulatory functions.

To date, insect resistance to Bt biopesticides and chemical pesticides has seriously threatened pest control in the field. The novel RNAi and CRISPR/Cas9 technologies are promising for insect pest control and resistance management in the near future. However, the RNAi and CRISPR/Cas9-based insect control strategies depend mainly on the selection of safe and efficient target genes, and many insect TFs are suitable candidate targets for lethal genes. For instance, mutations in gap genes such as *kni* can cause serious defects in embryos and impede their normal growth and development during *Drosophila* embryogenesis [17]. In *P. xylostella*, *abd-A* mutagenesis induced by the CRISPR/Cas9 system generated a heritable *abd-A* mutant phenotypes, resulting in severe abdominal morphological defects and significant lethality in the offspring [239]. Moreover, some important TFs, such as CncC and FoxA, have been found to be implicated in insect resistance [200,207], and these TFs can be used as insect lethal targets for pest management. Hence, the identification of these insect TFs will be conducive to developing both new species-specific biopesticides and next-generation transgenic crops combining Bt- and RNAi- or CRISPR/Cas9-based insect control technologies as a pivotal part of integrated pest management (IPM) programs [250].

With the development of high-throughput -omics techniques, the genome-wide identification of insect TFs is becoming easier, and subsequent TF studies will be performed at the genome level. Identifying a more comprehensive TF library in organisms is a major trend in the future. Considering the importance of versatile TFs in the transcriptional regulation of diverse insect physiological processes, undoubtedly, a growing body of entomologists will focus on studying insect TFs in the near future, and the vast range of genome resources and novel genetic methods will greatly propel progress of this area. Collectively, in-depth studies of insect TFs in the future will most likely provide new insights into the intracellular transcriptional regulation network of insects and even humans, which will have important potential for pest control in the field, and protection of human life and health.

## Figures and Tables

**Figure 1 ijms-19-03691-f001:**
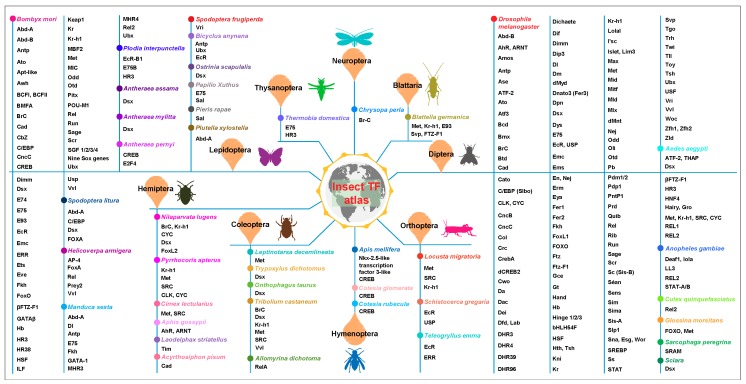
The insect transcription factor (TF) atlas, including the well-studied TFs and their related references collected in the review (See Appendix A for more detailed information about these TFs, and we apologize to researchers whose work could not be discussed and cited in the main text due to space limitations). As yet, diverse TFs have been documented in at least nine different insect orders including Diptera, Hemiptera, Lepidoptera, Coleoptera, Orthoptera, Thysanoptera, Blattaria, Neuroptera, and Hymenoptera. Different insect species are also denoted by colored circles on the vertical line.

**Figure 2 ijms-19-03691-f002:**
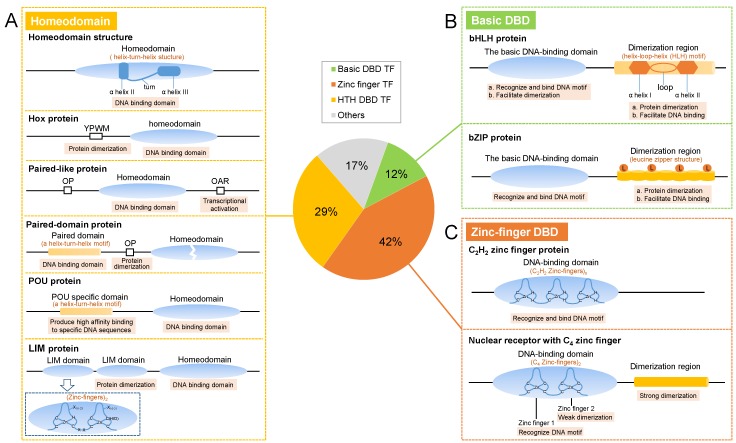
The main structures of insect TFs. The pie chart represents the statistical ratio of the *Drosophila* TFs in different superclasses based on Hammonds’ study [22]. (**A**) Structures of homeodomain TFs. Homeodomain: a typical homeodomain contains a short N-terminal arm facilitating DNA binding and four α-helices with a helix II-turn-helix III structure that is responsible for DNA binding and recognition; Hox protein: in addition to the homeodomain, a Hox protein typically contains a YPWM motif in the N-terminus mediating protein dimerization; Paired-like protein: in addition to the homeodomain, some paired-like proteins include an N-terminal octapeptide (OP) motif, and some contain a C-terminal OAR motif that can be involved in transcriptional activation; Paired-domain protein: paired-domain proteins carry an N-terminal paired box with a helix-turn-helix (HTH) structure that mediates DNA binding, and some proteins have a full-length or truncated homeodomain in the C-terminus, as well as an OP motif between the paired box and the homeodomain to mediate protein dimerization; POU (Pit-Oct-Unc) protein: POU proteins have a POU-specific domain with an HTH structure in the N-terminus that contributes to the generation of high-affinity DNA binding, and the homeodomain in the C-terminus is responsible for DNA binding; LIM protein: LIM proteins possess two LIM domains with zinc finger (ZF) structures upstream of the homeodomain that mainly mediate protein-protein interactions. (**B**) Structures of basic DBD TFs. The basic leucine zipper (bZIP) protein: bZIP proteins harbor a basic DNA-binding domain (DBD) for specific DNA recognition and binding, and a leucine zipper domain in the C-terminus for protein dimerization and DNA binding. The leucine zipper domain forms dextrorotatory α-helixes, and a leucine appears in the seventh position of every seven amino acids; thus, an adjacent leucine appears every two turns on the same side of the helix. The basic helix-loop-helix (bHLH) protein: bHLH proteins are composed of a basic DBD in the N-terminus, followed by an HLH domain. The basic DBD accounts for DNA motif recognition and binding and facilitates protein dimerization. In the HLH domain, two hydrophilic and lipophilic α-helices are separated by a loop to form an HLH structure mediating protein dimerization and contributing to DNA binding. (**C**) Structures of the ZF TFs. C_2_H_2_ ZF protein: the C_2_H_2_ ZF protein has multiple connected ZF DBDs. In the root of every ZF, two cysteines and two histidines link Zn^2+^ to form a tetrahedron. Nuclear receptor (NR) with C_4_ ZF: NR contains a C_4_-type ZF region as a DBD that consists of eight conserved cysteine residues coordinated with two Zn^2+^ to form two ZFs with a tetrahedral coordination structure. The first ZF provides DNA-binding specificity, and the second ZF has a weak dimerization interface, allowing for dimerization of the receptor molecule. In addition, a ligand-binding domain is typically found in the C-terminus and functions as the main dimerization region.

**Figure 3 ijms-19-03691-f003:**
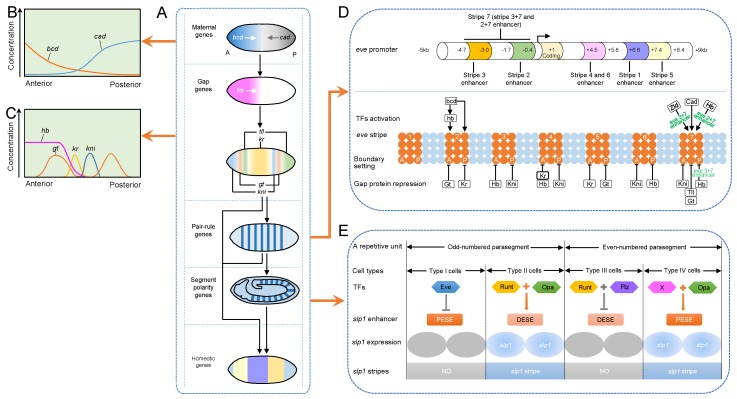
Establishment of the embryonic anterior–posterior (A-P) axis in *Drosophila*. The diagram is adapted from Gilbert’s works [40,41]. (**A**) Regulatory hierarchy of the formation of *Drosophila* A-P axis patterning. The maternal effect proteins Bicoid (Bcd) and Caudal (Cad) form a concentration gradient along the A-P axis and generate specific positional information to activate the expression of the gap gene *hunchback* (*Hb*). Hh further initiates the proper expression of other gap genes along the A-P axis. Gap proteins subsequently activate the expression of pair-rule genes, which form seven stripes perpendicular to the A-P axis and divide those discontinuous regions defined by the gap gene into body segments. The pair-rule proteins then regulate the expression of segment polarity genes in specific cells of each somite, and their 14 expression stripes establish the boundaries of parasegments. Finally, each segment is characterized by specifically expressed Hox genes. (**B**) The concentration gradient of the maternal effect proteins Bcd and Cad along the A-P axis in the early cleavage embryo. (**C**) The concentration changes in gap genes along the A-P axis. (**D**) Regulation of expression of the pair-rule gene *even-skipped* (*eve*) in seven stripes. The above region represents a partial promoter of the *eve* gene that contains five different enhancers responsible for the distinct stripes. The lower part illustrates how TFs regulate *eve* expression in different stripes. The black box represents the TF, the green characters indicate the enhancer of *eve*, and the orange circles display the cells expressing *eve*. The vertical bars with the letters “A” and “P” denote the anterior and posterior boundaries of the eve stripe, respectively, and the numbers show the number of *eve* bands. (**E**) Regulation of a repetitive unit of *sloppy paired 1* (*slp1*) stripe [42]. The 14-stripe patterning constitutes seven repetitive units, each containing odd-numbered and even-numbered parasegments. The odd-numbered parasegment consists of two types of cells: two type I cells in the posterior half do not express *slp1* and two type II cells in the posterior half that express *slp1*. The even-numbered parasegment also contains two types of cells: type III cells that do not express *slp1* in the first half and the latter type IV cells expressing *slp1*. The expression of *slp1* in different cell types is regulated by different pair-rule proteins in a specific combination to regulate the proximal early stripe element (PESE) or the distal early stripe element (DESE) of *slp1*. The colored hexagons indicate the pair-rule proteins, the orange quadrants represent the enhancer, the gray ovals exhibit the cells that do not express *slp1*, the blue ovals denote the cells expressing *slp1*, the gray rectangles with “No” show no *slp1* strips, and the blue rectangles represent *slp1* strips. TF abbreviations: Hb: Hunchback; Gt: Giant; Kr: Krüppel; Kni: Knirps; Tll: Tailess; Zld: Zelda; Opa: Odd-paired; Ftz: Fushi tarazu; X: represents an as yet unidentified Factor X.

**Figure 4 ijms-19-03691-f004:**
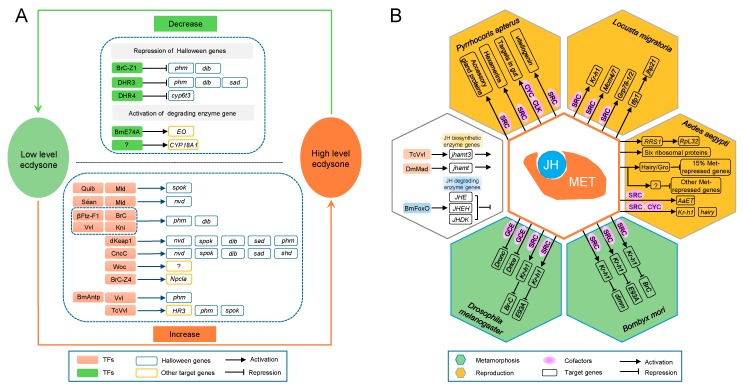
The 20-hydroxyecdysone (20E) and juvenile hormone (JH) signaling pathways regulating insect molting, metamorphosis, and reproduction. (**A**) Regulation of ecdysone titer by TFs. Changes in the ecdysone titers of insects are synergistically controlled by the synthesis and degradation of ecdysone. A low level of ecdysone promotes TFs to specifically regulate the Halloween genes in PG, thereby increasing steroidogenesis. A high level of ecdysone drives TFs to repress the expression of Halloween genes and activate the expression of degradative enzyme genes to directly degrade ecdysone, thus decreasing ecdysteroid titers. (**B**) Regulation of insect reproduction and metamorphosis by JH signaling pathway-related TFs. The white hexagon on the left shows TFs regulating the synthesis and degradation of JHs; the central hexagon represents JH binding to the JH receptor Met to regulate the transcription of downstream target genes; the orange hexagons display the JH/Methoprene-tolerant (Met) complex regulates downstream targets to control insect reproduction; the green hexagons indicate metamorphosis; and the purple gradient ellipses denote the cofactors of Met.

**Table 1 ijms-19-03691-t001:** *In silico* TF-binding site (TFBS) prediction by utilizing different TF databases and TFBS searching tools.

Names	Organisms	Websites	Descriptions	References
TRANSFAC	Eukaryotes	http://gene-regulation.com/	Partially commercial. License required to access some restricted areas.	[212]
JASPAR	Eukaryotes	http://jaspardev.genereg.net/	Contains a curated, non-redundant set of profiles, derived from published collections of experimentally defined eukaryotic TFBS.	[213]
DBD	Cellular organisms	http://www.transcriptionfactor.org/	Contains TF predictions of more than 1000 cellular organisms.	[212]
UniPROBE	Cellular organisms	http://the_brain.bwh.harvard.edu/uniprobe/	Contains DNA binding data for 638 non-redundant proteins and complexes from a diverse collection of organisms.	[214]
PlantTFDB	Plants	http://planttfdb.cbi.pku.edu.cn/	Contains 320,370 TFs from 165 plant species; enables regulation prediction and functional enrichment analyses.	[215]
LASAGNA-Search	Organisms	http://biogrid-lasagna.engr.uconn.edu/lasagna_search/	An integrated web tool for TFBS search and visualization.	[216]
PROMO	Organisms	http://alggen.lsi.upc.es/cgi-bin/promo_v3/promo/promoinit.cgi?dirDB=TF_8.3	A virtual laboratory for the identification of putative TFBS in DNA sequences from a species or groups of species of interest.	[217]
MatInspector	Organisms	http://www.genomatix.de/matinspector.html	A software tool that utilizes a large library of matrix descriptions for TFBS to locate matches in DNA sequences.	[218]
INSECT 2.0	Insects	http://bioinformatics.ibioba-mpsp-conicet.gov.ar/INSECT2/	A web-server for genome-wide *cis*-regulatory module prediction.	[219]

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
