# Peer review of "Insect Transcription Factors: A Landscape of Their Structures and Biological Functions in Drosophila and beyond"

_ijms, 2018, doi:10.3390/ijms19113691_

Round 1
Reviewer 1 Report
Authors present a very extensive, but largely uncritical review on the structure and function of transcription factors in Drosophila and other (non-model) insects. The manuscript mainly contains literature data from the last 20 years, with a focus on Drosophila. This is certainly of scientific value, especially because the applied methods are also discussed. In the method section, however, I miss examples of use in insects, I also miss two important references, the first significant paper on TF control in gene expression in insects by Harshman and James (1998) (Annu. Rev. Entomol.) and the recent paper by Parra et al. (2016) (Bioinformatics 32: 1229) on INSECT 2.0 (IN-silicio Search for Co-occurring Transcription factors).
Minor comments:
line 37: characteristics, not characterizes
line 52: existing
line 83: are required
line 114: cysteine residues
line 130: pancreas does not exist in insects. Do you mean the fat body?
line 182: TFs regulate
line 198: determine
line 242 and others: give a link to Table S1 for abbreviations of the TFs
line 261: Btd functions; to prevent
line 262: by inhibiting
line 275: say "Trachea and gland formation" because trachea are not a gland
line 388: the homeoprotein interacts...
line 422: to enhance
line 440: to promote
line 458: is 20E not also degraded? Better say "thus decreasing ecdysteroid titer"
line 459: TFs regulating...
line 461: the hexagons in Fig. 4B are orange, not yellow: JH/Met complex regulating...
line 463: cofactors
line 505: TFs regulate...
line 533: explain AMP (antimicrobial peptides) when first mentioned
line 567: give full name Culex, because first mentioned in text here
line 601 and others: say "hemimetabolous insects"
line 625: use C. here
line 663: give the species name Renilla in italics
line 751: induces
line 761: association study?
line 789: most likely provide...
Reference 121: volume 110 B or 110 C?
Author Response
#Reviewer 1
Comments and Suggestions for Authors
Comment 1: Authors present a very extensive, but largely uncritical review on the structure and function of transcription factors in Drosophila and other (non-model) insects. The manuscript mainly contains literature data from the last 20 years, with a focus on Drosophila. This is certainly of scientific value, especially because the applied methods are also discussed.
Response 1: We thank the reviewer for recognizing the relevance of the data presented and the positive comments, and we hope the responses and actions described below address the raised concerns by the reviewer.
Comment 2: In the method section, however, I miss examples of use in insects, I also miss two important references, the first significant paper on TF control in gene expression in insects by Harshman and James (1998) (Annu. Rev. Entomol.) and the recent paper by Parra et al. (2016) (Bioinformatics 32: 1229) on INSECT 2.0 (IN-silicio Search for Co-occurring Transcription factors).
Response 2: The reviewer gives good suggestions, and we have added the first significant paper on TF control in gene expression in insects by Harshman and James (1998) (Annu. Rev. Entomol.) in the Introduction section (Line 34). Moreover, we have added a new subtitle “4.1. TF-binding site (TFBS) prediction” to introduce the current popular TF databases and online softwares and websites to predict TFBS including INSECT 2.0 and have listed them in the newly added Table 1 (Lines 653-661).
Comment 3: Minor comments:
line 37: characteristics, not characterizes
line 52: existing
line 83: are required
line 114: cysteine residues
line 130: pancreas does not exist in insects. Do you mean the fat body?
line 182: TFs regulate
line 198: determine
line 242 and others: give a link to Table S1 for abbreviations of the TFs
line 261: Btd functions; to prevent
line 262: by inhibiting
line 275: say "Trachea and gland formation" because trachea are not a gland
line 388: the homeoprotein interacts...
line 422: to enhance
line 440: to promote
line 458: is 20E not also degraded? Better say "thus decreasing ecdysteroid titer"
line 459: TFs regulating...
line 461: the hexagons in Fig. 4B are orange, not yellow: JH/Met complex regulating...
line 463: cofactors
line 505: TFs regulate...
line 533: explain AMP (antimicrobial peptides) when first mentioned
line 567: give full name Culex, because first mentioned in text here
line 601 and others: say "hemimetabolous insects"
line 625: use C. here
line 663: give the species name Renilla in italics
line 751: induces
line 761: association study?
line 789: most likely provide...
Reference 121: volume 110 B or 110 C?
Response 3: We have revised these minor errors as the reviewer’s suggestion and highlighted them in yellow in the corresponding sections.
Reviewer 2 Report
General:
Well written and comprehensive, and a nice overview for non-geneticists. The authors do a good job covering Drosophila and combining that with a comparative approach with other model insect species, particularly in the section on "Biological Functions". The only place where the authors might consider some revision is to add more explicit discussion, and an example or two of mechanisms that connect their review to potential approaches to insecticides or other methods of pest control, since the main audience for this article appears to be scientists involved in agriculture.
Specific:
Abstract, line 16: Change "Noteworthy, as the development…." to "It is noteworthy that with the development of …."
Introduction, Figure 1: Omit this figure. It is already included in S1, it is too difficult to read and is purely descriptive. The remaining figures in the manuscript are much clearer and actually explain biological mechanisms.
Author Response
#Reviewer 2
Comments and Suggestions for Authors
General:
Comment 1: Well written and comprehensive, and a nice overview for non-geneticists. The authors do a good job covering Drosophila and combining that with a comparative approach with other model insect species, particularly in the section on "Biological Functions".
Response 1: We thank the reviewer for recognizing the relevance of the data presented and the positive comments, and we hope the responses and actions described below address the raised concerns by the reviewer.
Comment 2: The only place where the authors might consider some revision is to add more explicit discussion, and an example or two of mechanisms that connect their review to potential approaches to insecticides or other methods of pest control, since the main audience for this article appears to be scientists involved in agriculture.
Response 2: The reviewer gives good suggestions, and we have revised these sections (Lines 677-678, Lines 687-688, Lines 728-730, Lines 749-750, Lines 758-761, Lines 765-767, Lines 800-814) according to the reviewer’s suggestion.
Specific:
Comment 3: Abstract, line 16: Change "Noteworthy, as the development…." to "It is noteworthy that with the development of …."
Response 3: We have revised it as the reviewer’s suggestion.
Comment 4: Introduction, Figure 1: Omit this figure. It is already included in S1, it is too difficult to read and is purely descriptive. The remaining figures in the manuscript are much clearer and actually explain biological mechanisms.
Response 4: Although the content in Figure 1 is already included in Table S1, we really need a direct and vivid schematic diagram to let the readers quickly have an overall perspective about the current research status of TFs in different insect orders. Then, if the readers want to know some insect TFs in detail, they can see Table S1. This is the reason why we spent much time and energy to draw this figure. Therefore, in my opinion, we strongly hope the Figure 1 can be included in the paper to attract more potential readers thereby arousing their interest to further read and cite this paper. Of course, if the reviewer still holds his ground eventually, we will respect the final decision made by the Editor about whether Figure 1 should be retained after considering our explanation.